# Decontamination of Pathogenic and Spoilage Bacteria on Pork and Chicken Meat by Liquid Plasma Immersion

**DOI:** 10.3390/foods11121743

**Published:** 2022-06-14

**Authors:** Peeramas Sammanee, Phakamas Ngamsanga, Chalita Jainonthee, Vena Chupia, Choncharoen Sawangrat, Wichan Kerdjana, Kanninka Na Lampang, Tongkorn Meeyam, Duangporn Pichpol

**Affiliations:** 1Master’s Degree Program in Veterinary Science, Faculty of Veterinary Medicine, Chiang Mai University, Chiang Mai 50100, Thailand; peeramas_sam@cmu.ac.th; 2Department of Livestock Development, Ministry of Agriculture and Cooperatives, Bangkok 10400, Thailand; 3Veterinary Public Health and Food Safety Centre for Asia Pacific, Faculty of Veterinary Medicine, Chiang Mai University, Chiang Mai 50100, Thailand; phakamas.ng@cmu.ac.th (P.N.); chalita.j@cmu.ac.th (C.J.); tongkorn.meeyam@cmu.ac.th (T.M.); 4Center of Excellence in Veterinary Public Health, Faculty of Veterinary Medicine, Chiang Mai University, Chiang Mai 50100, Thailand; kannika.nalampang@cmu.ac.th; 5Department of Veterinary Biosciences and Veterinary Public Health, Faculty of Veterinary Medicine, Chiang Mai University, Chiang Mai 50100, Thailand; vena.ch@cmu.ac.th; 6Department of Industrial Engineering, Faculty of Engineering, Chiang Mai University, Chiang Mai 50200, Thailand; choncharoen@step.cmu.ac.th; 7Science and Technology Park, Chiang Mai University, Chiang Mai 50100, Thailand; wichan@step.cmu.ac.th

**Keywords:** foodborne pathogen, shelf-life, meat, cold plasma technology, decontamination

## Abstract

In this research, we aimed to reduce the bacterial loads of *Salmonella* Enteritidis, *Salmonella* Typhimurium, *Escherichia coli*, *Campylobacter jejuni*, *Staphylococcus aureus*, and *Pseudomonas aeruginosa* in pork and chicken meat with skin by applying cold plasma in a liquid state or liquid plasma. The results showed reductions in *S*. Enteritidis, *S*. Typhimurium, *E. coli*, and *C. jejuni* on the surface of pork and chicken meat after 15 min of liquid plasma treatment on days 0, 3, 7, and 10. However, the efficacy of the reduction in *S. aureus* was lower after day 3 of the experiment. Moreover, *P. aeruginosa* could not be inactivated under the same experimental conditions. The microbial decontamination with liquid plasma did not significantly reduce the microbial load, except for *C. jejuni*, compared with water immersion. When compared with a control group, the pH value and water activity of pork and chicken samples treated with liquid plasma were significantly different (*p* ≤ 0.05), with a downward trend that was similar to those of the control and water groups. Moreover, the redness (a*) and yellowness (b*) values (CIELAB) of the meat decreased. Although the liquid plasma group resulted in an increase in the lightness (L*) values of the pork samples, these values did not significantly change in the chicken samples. This study demonstrated the efficacy of liquid plasma at reducing *S*. Enteritidis, *S*. Typhimurium, *E. coli*, *C. jejuni*, and *S. aureus* on the surface of pork and chicken meat during three days of storage at 4–6 °C with minimal undesirable meat characteristics.

## 1. Introduction

The global population and the demand for meat consumption are growing; as a result, the production of food and meat is also increasing [1]. Global meat protein consumption is expected to rise by 14% at an estimated 138 million tons by 2030, compared with the period average of 121 million tons during 2018–2020 [2]. Bacterial contamination in food processing is a concern, since it is a cause of foodborne illness that can be harmful to the consumer [3]. The production of high-quality meat and meat products in the food industry faces a major challenge in reducing the risk of pathogens [4]. The global pork output is predicted to increase by 5% from October 2021 to 109.9 million tons in 2022, whereas global poultry meat production is expected to reach 100.8 million tons [5]. Additionally, the availability of pork and poultry meat protein is expected to increase by 13.1% and 17.8%, respectively, by 2030 [2]. Because of the increased population growth and the demands of meat consumption, the production of high-quality meat and meat products will also correspondingly increase, which will be a major challenge for the food industry considering the risk of pathogens [4].

During the slaughtering process, contamination with pathogens including *Salmonella* spp., *Escherichia coli*, *Campylobacter* spp., and spoilage bacteria can occur [6,7]. This can result in the contamination of food products with these bacteria [8,9]. 

The decontamination method during the slaughtering process is an essential condition to improve the quality of meat products. The decontamination method was divided into the following categories: (1) physical treatment, such as cold or hot water washing, steaming, and irradiation [10,11], and (2) chemical treatment such as organic acid, chlorine, ozone, trisodium phosphate, and cold plasma technology [12]. Physical modifications to meat and meat products are possible through the use of thermal decontamination procedures [13,14], freezing [15,16,17], irradiation [18,19,20], and trisodium phosphate [18,21]. In each method of decontamination, limitations can be found. For example, ozone can cause the rancidity of fat and muscle pigments in meat [22,23,24]. Additionally, the health and safety of meat handlers should be considered, including the risk of resistant acid bacteria such as *E. coli* O157:H7 [25], the corrosive effect of organic acid on meat industry equipment [26,27], and the risk of carcinogenic compounds as trihalomethanes (THMs) from working with chlorine [18,28], as several decontamination methods have been developed to mitigate the negative effects of existing methods. Thus, non-thermal (cold) plasma, particularly liquid plasma, is a novel decontamination technology that has the potential to aid in the elimination of pathogens in food or meat products and has been used in the meat business to overcome the limitations of conventional decontamination methods as previously mentioned. Liquid plasma is non-thermal plasma containing ions, atoms, and molecules, such as oxygen, ozone, nitrate, and nitrite. According to a study on non-thermal plasma in food processing, plasma can emit reactive oxygen and nitrogen species, which can eliminate harmful bacteria on meat surfaces [4,7,29]. Since liquid plasma can operate as a source of nitrite for meat products, a change in meat characteristics may also occur.

Chicken and pork are important protein sources and are in high demand with intensive production. Reducing the amount of bacterial contamination during the slaughtering process such as chilling in poultry or final washing in pig are categorized as crucial steps for the control point. Plasma technology has been applied with vegetables and fruits, as well as meat and meat products [30,31,32]. However, the application of liquid plasma in slaughtering and meat processing is limited. Therefore, the purpose of this research was to determine the efficiency of liquid plasma in reducing pathogenic and spoilage microorganisms in pork and chicken meat with skin and in modifying the quality of meat.

## 2. Materials and Methods

### 2.1. Experimental Design

Three groups of experiments were assigned in this study: liquid plasma, water, and the control treatment. In the liquid plasma group, meat samples were immersed in 500 mL of 60 ppm of H_2_O_2_ for 15 min at 25.5 °C. The water group was treated with 500 mL of sterile distilled water for 15 min. The control group consisted of samples that had not been exposed to any solution. Each strain of the bacteria was replicated in seven samples in pork and chicken with skin per group.

### 2.2. Preparation of Meat 

The meat samples in this study were a pork belly with skin, composed of multiple muscles such cutaneous trunci, latissimus dorsi, pertoralis profundus, rectus abdominis, internal and external abdominal oblique, and fat tissues between these muscles [33], and a chicken breast meat with skin including the pectoralis major muscle part. We purchased the meat with skin from GMP-certified slaughterhouses that had total bacteria and *Enterobacteriaceae* counts of fewer than 4.0 log CFU/cm^2^ and that tested negative for *Salmonella* and *Campylobacter*. Meat samples from the same farm and batch of production were collected to control the color, pH, and core temperature of the meat during the cutting process. The cutting size of the meat was approximately 15 × 15 × 4 cm. The average weights were 315 ± 57 g of pork and 296 ± 40 g of chicken meat. The storage temperature of the meat was −18 °C.

### 2.3. Liquid Plasma Preparation

The liquid plasma was prepared by Science and Technology Park, Chiang Mai University, Thailand. It was generated by a pinhole plasma jet under atmospheric pressure. Argon gas was used as the carrier gas. The power supply was 15 kv with a 125-watt power density using the alternating current electricity of a neon transformer, and the frequency was 50 Hz. The concentration of hydrogen peroxide (H_2_O_2_) in the liquid plasma was 59.29 ± 4.46 ppm. A spectrophotometer was used to determine the concentration of H_2_O_2_ (Perkin Elmer Lambda 25, Norwalk, CT, USA) [34]. Following the generation process, the liquid plasma was directly used in the experiment.

### 2.4. Bacterial Strains

The *Salmonella* Enteritidis nalidixic acid-resistant strain, *S*. Typhimurium (wildtype strain), *Campylobacter jejuni* ATCC 33560, *Escherichia coli* ATCC 25922, *Staphylococcus aureus* ATCC 25923, and *Pseudomonas aeruginosa* ATCC 27853 were recovered from −80 °C stock on 5% sheep blood agar (Oxoid, Hampshire, UK) and incubated at an optimal temperature and atmosphere. The quality control of the bacterial concentration was performed using the drop-plating technique on selective media for all batches of the bacterial suspension. 

### 2.5. Inoculation of Bacterial Suspension

All of the meat samples were packed in sterile plastic sealed bags. A total of 4 log CFU/mL of the bacterial suspension was inoculated and spread thoroughly on the samples’ surfaces using a sterile glass rod. They were left at room temperature for 15 min to allow for the bacterial attachment before further experimental processes [35,36].

### 2.6. Microbiological Analysis

The samples were refrigerated in the sterile plastic sealed bags at 4–6 °C to perform the microbial analysis four separate times on days 0, 3, 7, and 10.

#### 2.6.1. Detection and Enumeration of Salmonella

The samples were cut to 25 g in sterile stomacher bags and 225 mL of sterile buffered peptone water (BPW; Merck, Darmstadt, Germany) was added. The sample was then homogenized with a stomacher (Interscience, Saint-Nom-la-Brétèche, France) for 2 min before being incubated at 37 °C for 24 h. A total of 100 μL of the overnight-incubated suspension was then pipetted onto the modified semisolid Rappaport–Vassiliadis (MSRV; Merck, Darmstadt, Germany) agar with three equally spaced spots and incubated at 41.5 °C for 24 h. *Salmonella* growth on the MSRV agar appeared to be opaque. One loopful (10 μL) of the furthest point of the opaque growth was picked up and steaked on brilliant green phenol red lactose sucrose agar (BPLS; Merck, Darmstadt, Germany) and xylose lysine deoxycholate agar (XLD; Merck, Darmstadt, Germany) and incubated at 37 °C for 24 h. The typical *S. enterica* cultured on XLD agar exhibited a clear pink appearance with or without a black center on pink agar. Additionally, the red zone surrounding the pink *Salmonella* colony was observed on BPLS agar. The typical colonies of *Salmonella* spp. were then selected for biochemical and serology confirmation [37].

To count *Salmonella*, a cut of 25 g of meat sample was added with 225 mL of sterile BPW (Merck, Darmstadt, Germany), and then homogenized with a stomacher for 2 min. A ten-fold serial dilution of the homogenate was performed before being dropped onto XLD agar (Merck, Darmstadt, Germany) and BPLS agar (Merck, Darmstadt, Germany) and incubated at 37 °C for 24 h. The typical colonies were counted and tested following the ISO 6579 [37].

#### 2.6.2. Enumeration of *E. coli*

The drop-plating technique was performed to enumerate *E. coli* on *E. coli* direct agar (Fluorocult^®^ ECD agar, Merck, Darmstadt, Germany). In a sterile stomacher bag, a 25 g meat sample was placed and 225 mL of the sterile BPW (Merck, Darmstadt, Germany) was added. Then, the sample was homogenized using a stomacher for 2 min and a ten-fold dilution was performed. A 50 μL sample of each dilution was dropped on ECD agar, and then incubated at 44 °C for 24 h in an aerobic condition. The round, turbid blue–white illuminated colonies were observed under a UV lamp (366 nm) and indole production was tested to identify the bacteria [38,39].

#### 2.6.3. Detection and Enumeration of Campylobacter

A 25 g meat sample was mixed with 225 mL Bolton broth (Oxoid, Hampshire, UK and homogenized using a stomacher for 2 min, and then incubated at 37 °C for 4–6 h under microaerobic conditions (CampyGen, Oxoid, Hampshire, UK) and at 41.5 °C for 48 h. A loopful of the suspension was inoculated on a modified charcoal cefoperazone deoxycholate (mCCD; Oxoid, Hampshire, UK) agar and Preston agar (Oxoid, Hampshire, UK) and incubated at 41.5 °C for 48 h. To identify *Campylobacter*, the typical flat and moist grayish colonies, frequently with a metallic sheen, were observed on mCCD agar and the moist, gray, flat spreading colonies were observed on Preston agar [40]. The suspected colonies were streaked on a Columbia blood agar (CBA; Oxoid, Hampshire, UK) supplemented with 5% (*v/v*) sterile defibrinated sheep blood (Clinag, Bangkok, Thailand) and incubated under microaerobic conditions as described at 41.5 °C for 48 h. The pure cultures were then confirmed by morphology, motility, and oxidase tests [41,42], and multiplex PCR for genus and species confirmation followed the primers and condition in Denis’ study [43,44]. For *Campylobacter* enumeration, the homogenate of 25 g and 225 mL of the diluent underwent ten-fold dilution, and 50 μL of the sample was dropped onto mCCD agar and incubated at 41.5 °C for 48 h under microaerobic conditions [42]. 

#### 2.6.4. Detection and Enumeration of *S. aureus*

The sample was cut to 25 g and placed in a sterile stomacher bag, and then 225 mL of the maximum recovery diluent (MRD; Merck, Darmstadt, Germany) was added and the sample was homogenized using a stomacher for 2 min. The homogenate underwent ten-fold dilution and was dropped on Baird–Parker agar (BPA; Merck, Darmstadt, Germany) before being incubated at 37 °C for 48 h. The typical black, convex, shiny colonies, with a diameter of 1–1.5 mm, were observed on BPA. To identify *S. aureus*, the clear halos surrounding colonies appeared after 48 h of incubation and tested positive with a coagulase test [45].

#### 2.6.5. Detection and Enumeration of the Pseudomonad

The sample was cut to 25 g and placed in a sterile stomacher bag. A total of 225 mL of MRD was added, and then homogenized for 2 min with a stomacher. The homogenate underwent ten-fold dilution and drop plating was performed to count the bacteria on glutamate starch phenol red agar (GSP; Merck, Darmstadt, Germany). The homogenate was then incubated at 30 °C for 48 h. The typical large, blue-violet, and red-violet surrounding colonies that were 2–3 mm in diameter were observed on GSP agar and tested positive with oxidase [46,47].

### 2.7. Evaluation of Meat Quality

#### 2.7.1. Meat Color

The evaluation of the meat color was performed on days 0, 3, 7, and 10 at the skin site and meat site in both types of meat samples (and side part with a fat layer in the pork sample) using a CIE colorimeter (MiniScan, Hunter Associates Laboratory, Inc., Reston, VA, USA) as shown in Figure 1, and then the values were recorded as averages [48]. The parameters of the CIELAB were as follows:L*: lightness from black to white,L0: black,L100: white,a*: red–green index,a +: red,a −: green,b*: yellow–blue index,b +: yellow,b −: blue.

#### 2.7.2. Water Activity (a_w_) of Meat 

The samples from days 0, 3, 7, and 10 were cut into small pieces and placed in a sample container. The samples were then placed in a water activity meter (AquaLab^®^, Washington, DC, USA) for evaluation, and the values of a_w_ and the temperature of the samples were then recorded [49].

#### 2.7.3. pH of Meat

The pH value of the samples at nine positions were measured at the surface and inside of the meat using a pH meter (CyberScan^®^ pH 310 series, Crescent, Singapore) on days 0, 3, 7, and 10, and the values were then recorded.

### 2.8. Statistical Analysis

The data of the microbial load from all experiments were expressed as log CFU/g. A two-way repeated ANOVA was used to compare the microbial load, color, pH, and water activity from each treatment group (liquid plasma, water immersion, and control groups) and the day of the experiment (days 0, 3, 7, and 10). A Bonferroni test as a multiple comparison with a confidence level of *p* ≤ 0.05 was used to analyze the significant differences between the mean values of the data from each of the treatment groups.

## 3. Results

### 3.1. Effect of Liquid Plasma on Inactivation of Pathogenic Bacteria

The immersion of the pork and chicken samples in liquid plasma (60 ppm of H_2_O_2_) for 15 min with the artificial inoculation of *S.* Enteritidis, *S.* Typhimurium, *E. coli*, and *C. jejuni* resulted in a reduction in the microbial loads. The reduction in the microbial loads of all of the abovementioned bacteria from day 0 to 10 in the liquid plasma group and water immersion group, compared with the control group, showed that the reduction of the liquid plasma group was not higher than that in the water group, except for *C. jejuni*, with reduction levels of 1.03 ± 0.99 log CFU/g in pork (Table 1), and 1.39 ± 1.10 log CFU/g in chicken meat (Table 2). The reduction of *C. jejuni* was reduced significantly when compared with the reduction of *S.* Typhimurium within the liquid plasma group (*p* ≤ 0.05), as shown in Table 1. However, the other bacterial reduction results in the water and plasma treatments were not significantly different in the pork and chicken samples (Table 1 and Table 3). The findings of the microbiological investigation of *S. aureus* after day 3 of the decontamination experiment in the pork and chicken samples indicated that the microbial loads started to increase. Moreover, the amount of *P. aeruginosa* in the pork and chicken samples on day 10 was overgrowth. The mean differences in the microbial loads between all treatment groups at each time point were illustrated in Figure 2 and Table 3 for the pork samples, and in Figure 3 and Table 4 for the chicken samples.

### 3.2. pH Values of Meat

The pH values on the surface and inside of the pork and the chicken samples after the application of liquid plasma with 60 ppm of H_2_O_2_ for 15 min on days 0, 3, 7, and 10 are illustrated in Table 5, Table 6, Table 7, and Table 8, respectively. There were significant decreases in the pH values in the liquid plasma group at days 0 and 10. In addition, comparisons of the pH values among each treatment group showed that most of the averages of the pH values were significantly different at each time point, except for the averages of the pH values inside the chicken samples on day 3 (Table 8).

### 3.3. Color of Meat 

The CIELAB color values in the samples of pork belly and chicken meat with skin treated with liquid plasma for 15 min are shown in Table 9 and Table 10, respectively. The results show that the L* value of all parts of the pork samples increased with each time point, when compared with the control and water groups. On the other hand, the L* values of all parts of the chicken samples were not significantly different. Additionally, a* values of the samples of pork side parts with a fat layer as well as the skin parts of chicken samples in the liquid plasma group were significantly lower than the control and water treatments. Furthermore, the b* values of skin parts, pork side parts with a fat layer, and the meat parts of the chicken meat samples after the application of the liquid plasma were significantly lower than the control and water treatments. However, the b* values of the red meat parts of the pork samples in the liquid plasma group were significantly higher than the water immersion at day 10. 

### 3.4. Water Activity of Meat

The a_w_ in the pork samples treated with the liquid plasma slightly changed and tended to decrease at each time point. However, the a_w_ values of the pork and chicken samples treated with the liquid plasma remained similar to the control and water treatments, as shown in Table 11. 

## 4. Discussion

In the decontamination experiment, the application of the liquid plasma with 60 ppm of H_2_O_2_ for 15 min could reduce the microbial loads of *S.* Enteritidis, *S.* Typhimurium, and *E. coli* of the chicken meat’s surface, reaching values lower than those with decontamination with water. As of day 7 of the experiment, the chicken meat developed an unpleasant odor, resulting in an increase in the microbial loads of all strains of the bacteria in this study, leading to contamination from day 7 to day 10. The findings revealed that the chicken samples with the artificial inoculation of the bacteria were not able to be kept at the refrigerator temperature of 4–6 °C for more than 7 days. Moreover, in an experiment of the pork samples, the microbial loads of *S*. Enteritidis, *S*. Typhimurium, and *E. coli* decreased, but not significantly, when compared to the water treatment. Gas plasma in free-radical form, such as hydroxyl and hydroperoxyl, is generated under the stimulating energy to excite the gas molecules. These molecules perform charged particles in the form of free radicals. Their microbicidal ability is the interaction with essential cell components and destruction of the metabolism of microorganisms [50]. The water droplets in the liquid plasma process can suppress the radical deterioration reaction of hydrogen peroxide, and may enhance plasma efficiency [51]. Argon plasma is one of the most promising options for producing a high concentration of H_2_O_2_, since the hydroxyl dissolved in the argon plasma and hydroxyl radicals are the principal source of H_2_O_2_ formation that will be transferred to treated liquid or plasma-activated water (PAW) and contribute to the inactivation of bacteria [52]. In a study by Laurita et al., the H_2_O_2_ concentration in DBD air plasma-triggered PAW was analyzed and observed to be 200 μM. However, after 25 min of post-discharge kinetics, H_2_O_2_ decomposition in PAW at various intervals, the H_2_O_2_ concentration reduced to approximately 110 μM [53]. A study by Lee involved applying dielectric barrier discharge (DBD) plasma treatment to chicken breasts for 10 min, leading to a decrease in *E. coli* and *S.* Typhimurium of 2.73 and 2.71 log CFU/g, respectively [54]. In Fernandez and Thompson’s study, it was suggested that the efficacy of atmospheric cold plasma (ACP) for bacterial inactivation diminished when surface convolutions such as food surfaces or attachment sites increased [55]. When ACP was used to decontaminate eggshells inoculated with *S.* Enteritidis and *S*. Typhimurium, it was discovered that contact times longer than 1 h were necessary to inactivate all of the abovementioned bacteria [56]. Additionally, Mostafa’s study found that when chicken carcasses were immersed in water containing 0.1% of H_2_O_2_, the total colony counts, coliform counts, and *S. aureus* counts decreased when compared to untreated samples, indicating that adding H_2_O_2_ to the water used for chilling carcasses could help reduce bacterial contamination [57]. Furthermore, the study by Kim found that exposing DBD plasma to He and He plus O_2_ reduced the loads of *E. coli* by 0.26 and 0.50 log CFU/g, respectively, for 5 min as well as by 0.34 and 0.55 log CFU/g, respectively, for 10 min of application [58]. As a result, the efficiency of bacterial reduction in food and on meat surfaces was correlated with surface convolution, contact time, chemical substances, and input gas.

During 10 days of storage, the loads of *C. jejuni* on the surfaces of pork and chicken meat with skin were decreased by approximately 1 log CFU/g compared to the water and control groups. The results indicated that samples treated with liquid plasma containing 60 ppm H_2_O_2_ for 15 min might significantly reduce *C. jejuni* microbial loads when maintained at refrigerator temperature (4–6 °C) for the recommended storage time for each type of meat. *Campylobacter* spp. grows in a microaerobic or limited oxygen atmosphere. Reactive oxygen species, such as hydrogen peroxide, super oxide anion, and the hydroxyl radicals, are produced during aerobic respiration [59]. During the cold plasma production process, electrons and ions are generated. These particles include reactive oxygen species, which have antimicrobial properties [7,59]. As a result, *C. jejuni* can be inactivated following treatment with liquid plasma containing 60 ppm H_2_O_2_.

In addition, the microbial loads of pork and chicken samples artificially contaminated with *S. aureus* in all treatment groups were low on day 0, but increased after day 3. Pork and chicken meat treated with liquid plasma may not be suitable for storage in the refrigerator for more than three days at a temperature of 4–6 °C in this experimental environment. A study by Shen found that lowering the temperature enhanced the bactericidal effectiveness of PAW at −80 °C > −20 °C > 4 °C > 25 °C, respectively. Compared to PAW stored at 25 °C, 4 °C, and 20 °C, which exhibited a 0.2–2 log reduction in *S. aureus* for over 30 days after PAW generation, PAW stored at −80 °C had the greatest antibacterial activity, resulting in a 3–4-log reduction [60]. According to our study, the decontamination experiment was conducted at room temperature (25–30 °C), which could have had less antibacterial activity than those conducted at the lower temperatures. The effectiveness of the ACP treatment depended on the type of bacteria. Biofilms are extremely complex biomaterials that vary depending on the type and strain of bacteria [61]. After 60 sec of ACP treatment, colony counts demonstrated that *E. coli* biofilm populations were reduced to undetectable levels, although *S. aureus* biofilm populations were less affected [62].

Similarly, *P. aeruginosa* decreased on day 0 in the experiment with liquid plasma decontamination and did not appear to be inactivated by days 3–10. In the study of Ziuzina, the colony counts of the *P. aeruginosa* biofilm using DBD plasma were reduced by 5.44 log CFU/mL after 60 sec of treatment. Conversely, after 120 and 300 sec, the inactivation rate was observed to be lower [63]. Additionally, Patenall’s work found that ACP treatment reduced the *P. aeruginosa* biofilm at 0 and 4 h, with the reduction levels of 5 log CFU/mL. However, after the exposure time of 12, 20, and 24 h, the biofilm of *P. aeruginosa* had only reduced by 1–2 log CFU/mL [64]. 

In our trials, the artificial inoculation had a bacterial concentration of 4 log CFU/mL, which was a relatively high level of initial load. Furthermore, maintaining the stability of reactive oxygen species such as hydrogen peroxide during storage is essential to antimicrobial properties; the decrease of hydrogen peroxide over storage time at 25 °C, 4 °C, and −20 °C has been suggested to affect microbial inactivation by PAW compared to PAW stored at −80 °C [60]. In addition, our experiment was performed at room temperature (25–30 °C), affecting the antimicrobial activity of liquid plasma as well as the H_2_O_2_ concentration, which could decrease over a period of time. Since the microbial reduction using liquid plasma was low, considering a dose higher than 60 ppm hydrogen peroxide (H_2_O_2_) in the liquid plasma could enhance the decontamination performance. However, this may lead to a bleaching effect that would be undesirable for retailers and consumers [65]. The results show a reduction in antimicrobial activity that might lead to the liquid plasma treatment performing similarly to the water treatment, therefore potentially compromising the decontamination efficiency of the liquid plasma. A combination of liquid plasma treatment and good hygienic practices in the slaughtering process could be more efficient in lowering the bacterial load in meat. 

Additionally, the pH values in the liquid plasma treatment were relatively low, but stayed within the range of 5.4–6.2 for the normal pork and 5.6–6.4 for the normal chicken meat [66]. According to the findings, liquid plasma with 60 ppm of H_2_O_2_ could be an effective decontamination procedure for pork and chicken with skin surfaces. On the other hand, in the study of Kim, the pH value of the DBD plasma-treated pork loins was approximately 5.3, which was significantly lower than the untreated meat samples [58]. Moutiq’s study reported that after being treated with DBD plasma for the duration of each treatment, the pH values of the chicken breast samples decreased overall (1, 3, and 5 min). The decrease in the pH value could be caused by a deficiency in acidogenic molecules such as nitric and nitrous acid, which frequently form in the gaseous phase and are released into the water-covering muscle [67]. However, variations in the plasma type, process gas, decontamination technique, moisture of the surface material, and muscle type can all have an effect on the pH value following cold-plasma application [58,67]. 

The color and overall appearance of meat have a greater effect on customer acceptance before consumption than flavor and texture [68]. According to our findings, the liquid plasma treatment had an effect on the appearance of the pork and chicken meat, such as paleness. When comparing the L* (lightness) value (CIELAB) of all parts of the pork samples treated with liquid plasma using 60 ppm of H_2_O_2_ for 15 min to the control and water groups, the L* values of all parts of the pork samples increased at each time point. Conversely, the L* values of the chicken samples did not change significantly. Peroxidation is associated with the coloration of raw meat, since oxidative reactions have a pronounced effect on the concentration and chemical state of the heme proteins, myoglobin, and hemoglobin in muscle. The brown metmyoglobin form is produced and accumulated in muscles as a result of myoglobin oxidation to metmyoglobin, impacting the meat’s color [69]. Mulder’s study reported that immersing chicken carcasses in 0.17%, 0.5%, and 1% of H_2_O_2_ for 10 min bleached and bloated the carcasses [70]. However, these adverse effects vanished after one day of storage at 1 °C, which may be attributed to the natural catalase activity releasing oxygen from the skin and blood following treatment [70]. Additionally, Zhuang’s study discovered that after five days of post-treatment storage, the L* value of chicken breast treated with in-package cold plasma was dramatically raised, resulting in paler breast meat [31].

Moreover, on the side part with a fat layer, the a* (redness) values of the pork treated with liquid plasma were significantly lower than the control and water groups. In addition, after applying the liquid plasma to the skin and side part with a fat layer of the pork samples, the b* (yellowness) value of the skin and side part with a fat layer was lower than the control and water groups. However, on day 10, the b* values of the red meat in the liquid plasma group were significantly higher than those of the water immersion group. A color evaluation of pork loins following treatment with DBD plasma revealed that the L* values significantly decreased, the a* values slightly decreased, and the b* values remained unchanged [58]. The a* values of the chicken samples’ skin in the liquid plasma group were significantly lower than the control and water groups. Furthermore, the b* values of the meat part were significantly lower than the control and water groups. The L*, a*, and b* values remained close to the untreated samples in the study of the in-package decontamination of chicken breast using ACP plasma [67]. According to Lee’s study, the L* and b* values of the chicken breast treated with flexible thin-layer DBD plasma increased but the a* values decreased, which correlated with an increase in exposure time [54]. 

In this study, the water activity of the pork samples treated with liquid plasma was slightly lower, but remained close to the untreated sample. In contrast, the water activity of the sample of chicken with skin treated with liquid plasma was similar to that of the control and water groups. The general water activity value of raw meat was 0.98 or above [71]. Additional cold plasma investigations are required to determine the components and procedures responsible for lowering water activity in order to preserve meat or inactivate bacteria.

## 5. Conclusions

In this study, cold plasma technology in liquid form or liquid plasma with 60 ppm H_2_O_2_ could effectively reduce the amount of *S*. Enteritidis, *S*. Typhimurium, *E. coli*, and *C. jejuni* on the surface of pork and chicken meat. However, the treatment had a limited effect on the reduction of *S. aureus* and *P. aeruginosa*. This treatment appeared to have a negligible effect on the color, pH value, and water activity of the meat. This method may represent a novel technology for use in the pig and chicken slaughtering process.

## Figures and Tables

**Figure 1 foods-11-01743-f001:**
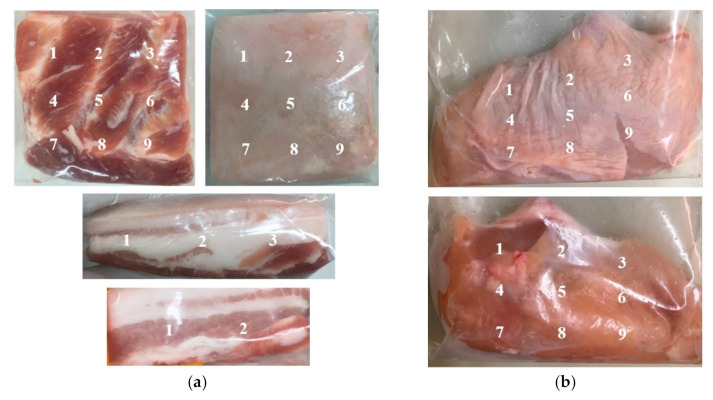
Sites for the color evaluation of pork with skin (**a**) and sites for the color evaluation of chicken meat with skin (**b**).

**Figure 2 foods-11-01743-f002:**
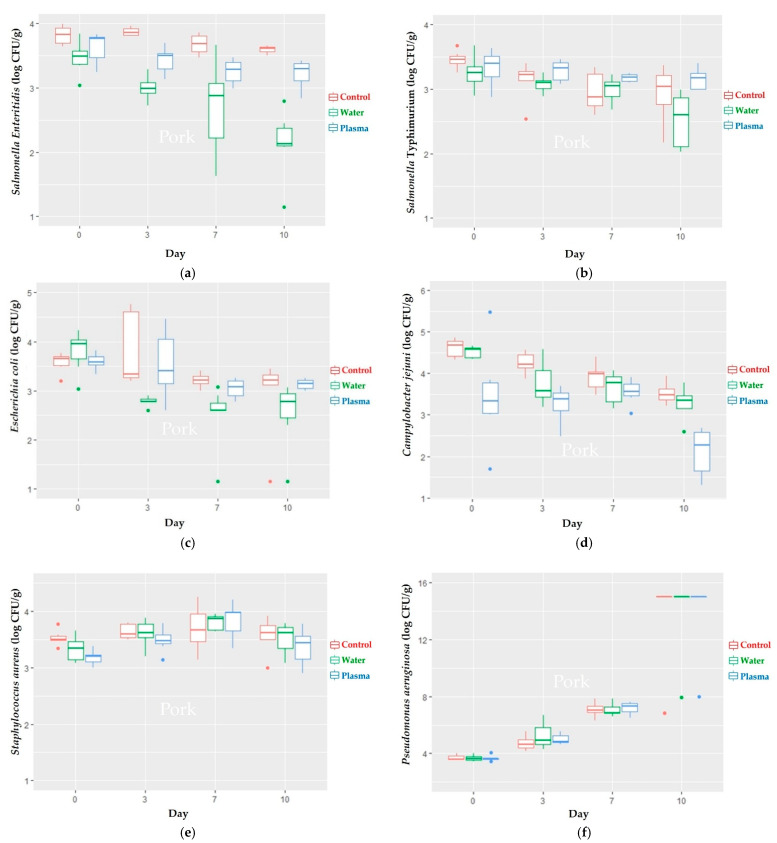
Microbial loads of (**a**) *S.* Enteritidis, (**b**) *S.* Typhimurium, (**c**) *E. coli*, (**d**) *C. jejuni*, (**e**) *S. aureus*, and (**f**) *P. aeruginosa* of pork samples at day 0 of the experiment and after being stored at 4–6 °C for 10 days (*p* ≤ 0.05).

**Figure 3 foods-11-01743-f003:**
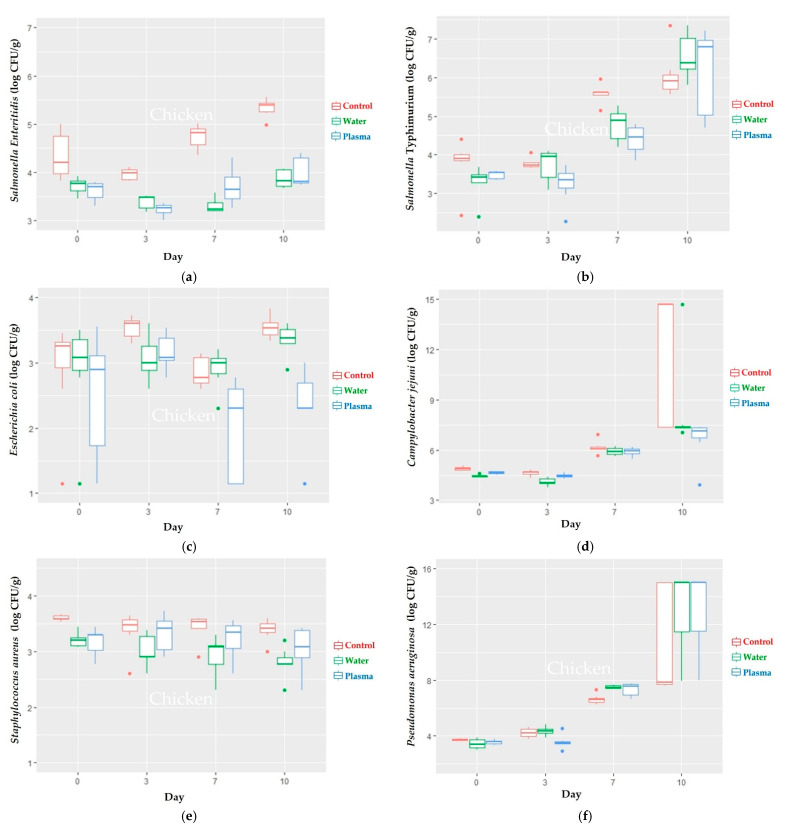
Microbial loads of (**a**) *S*. Enteritidis, (**b**) *S*. Typhimurium, (**c**) *E. coli*, (**d**) *C. jejuni*, (**e**) *S. aureus*, and (**f**) *P. aeruginosa* of chicken samples at day 0 of the experiment and after being stored at 4–6 °C for 10 days (*p* ≤ 0.05).

**Table 1 foods-11-01743-t001:** Reduction in the bacteria on pork.

Bacterial Strains	Control	Reduction (log CFU/g ± SD) *	
		Water	Liquid Plasma
*Salmonella* Enteritidis	3.75 ± 0.17	0.93 ± 0.67	0.37 ± 0.25
*Salmonella* Typhimurium	3.31 ± 0.41	0.20 ± 0.48	0.08 ± 0.19 ^a^
*Escherichia coli*	3.41 ± 0.64	0.50 ± 0.72	0.08 ± 0.43
*Campylobacter jejuni*	4.07 ± 0.48	0.27 ± 0.57	1.03 ± 0.99 ^a^
*Staphylococcus aureus*	3.62 ± 0.26	0.05 ± 0.27	0.15 ± 0.34
*Pseudomonas aeruginosa*	7.34 ± 4.28	−0.67 ± 4.23 **	−0.13 ± 4.25 **

^a^: The same lowercase letters within a column indicate that the values are significantly different at *p* ≤ 0.05. * Reduction is the difference of the mean (log CFU/g) of the bacteria between the initial load (control or untreated samples) and each treatment. ** Reduction (log CFU/g ± SD) is negative due to an increase in the number of microorganisms after each treatment.

**Table 2 foods-11-01743-t002:** Reduction in the bacteria in chicken meat.

Bacterial Strains	Control	Reduction (log CFU/g ± SD) *	
		Water	Liquid Plasma
*Salmonella* Enteritidis	4.59 ± 0.58	1.04 ± 0.33	1.11 ± 0.52
*Salmonella* Typhimurium	4.79 ± 1.14	0.20 ± 1.36	0.49 ± 1.30
*Escherichia coli*	3.21 ± 0.53	0.15 ± 0.49	0.72 ± 0.80
*Campylobacter jejuni*	6.81 ± 3.40	1.09 ± 2.16	1.39 ± 1.10
*Staphylococcus aureus*	3.45 ± 0.25	0.46 ± 0.29	0.26 ± 0.33
*Pseudomonas aeruginosa*	6.36 ± 3.41	−0.79 ± 4.14 **	−0.58 ± 4.26 **

* Reduction is the difference of the mean (log CFU/g) of the bacteria between the initial load (control or untreated samples) and each treatment. ** Reduction (log CFU/g ± SD) is negative due to an increase in the number of microorganisms after the treatment.

**Table 3 foods-11-01743-t003:** Means of the microbial loads (log CFU/g ± SD) of pork samples at days 0, 3, 7, and 10 after being stored under refrigerated conditions.

Bacterial Strains	Group	Day 0	Day 3	Day 7	Day 10
*S.* Enteritidis	Control	3.86 ± 0.17	3.88 ± 0.06 ^a^	3.68 ± 0.16	3.60 ± 0.05 ^a^
	Water	3.47 ± 0.25 ^a,b^	3.00 ± 0.18 ^a,c^	2.68 ± 0.77	2.15 ± 0.51 ^b,c^
	Liquid plasma	3.62 ± 0.23 ^a^	3.43 ± 0.20	3.25 ± 0.19 ^a^	3.22 ± 0.21
*S.* Typhimurium	Control	3.46 ± 0.13 ^a^	3.14 ± 0.28	2.97 ± 0.30	2.94 ± 0.42 ^a^
	Water	3.25 ± 0.25	3.08 ± 0.12 ^a^	3.00 ± 0.19	2.51 ± 0.42 ^a^
	Liquid plasma	3.33 ± 0.27	3.28 ± 0.16	3.17 ± 0.06	3.15 ± 0.16
*E. coli*	Control	3.58 ± 0.19	3.87 ± 0.74	3.22 ± 0.14	2.96 ± 0.81
	Water	3.80 ± 0.41 ^a,b,c^	2.79 ± 0.10 ^a^	2.50 ± 0.63 ^b^	2.54 ± 0.67 ^c^
	Liquid plasma	3.60 ± 0.16 ^a,b^	3.55 ± 0.69	3.05 ± 0.19 ^b^	3.13 ± 0.11 ^b^
*C. jejuni*	Control	4.61 ± 0.22 ^a,b^	4.26 ± 0.24 ^c^	3.90 ± 0.31 ^a,c^	3.52 ± 0.25 ^b,c^
	Water	4.52 ± 0.13 ^a,b,c^	3.77 ± 0.48 ^a^	3.64 ± 0.37 ^b^	3.28 ± 0.37 ^c^
	Liquid plasma	3.45 ± 1.14	3.26 ± 0.44 ^a^	3.56 ± 0.29 ^b^	1.88 ± 0.81 ^a,b^
*S. aureus*	Control	3.53 ± 0.13	3.65 ± 0.13	3.70 ± 0.39	3.58 ± 0.30
	Water	3.33 ± 0.22 ^a^	3.62 ± 0.23	3.80 ± 0.13 ^a^	3.51 ± 0.28
	Liquid plasma	3.18 ± 0.12 ^a^	3.49 ± 0.20 ^b^	3.83 ± 0.30 ^a,b^	3.37 ± 0.32
*P. aeruginosa*	Control	3.72 ± 0.17 ^a^	4.72 ± 0.49 ^a^	7.09 ± 0.49 ^a^	>8.00 ± 0.00 ^a^
	Water	3.68 ± 0.20 ^a^	5.27 ± 0.87 ^a^	7.08 ± 0.46 ^a^	>8.00 ± 0.00 ^a^
	Liquid plasma	3.67 ± 0.20 ^a^	5.01 ± 0.35 ^a^	7.21 ± 0.45 ^a^	>8.00 ± 0.00 ^a^

^a,b,c^: The same lowercase letters within a row indicate that the values are significantly different at *p* ≤ 0.05.

**Table 4 foods-11-01743-t004:** Means of the microbial loads (log CFU/g ± SD) of chicken samples at days 0, 3, 7, and 10 after being stored under refrigerated conditions.

Bacterial Strains	Group	Day 0	Day 3	Day 7	Day 10
*S.* Enteritidis	Control	4.35 ± 0.48 ^a^	3.96 ± 0.11 ^b,c^	4.73 ± 0.26 ^b,d^	5.33 ± 0.19 ^a,c,d^
	Water	3.72 ± 0.18	3.39 ± 0.14 ^a^	3.21 ± 0.30 ^b^	3.87 ± 0.18 ^a,b^
	Liquid plasma	3.62 ± 0.19 ^a^	3.05 ± 0.25 ^a^	3.71 ± 0.37	3.55 ± 0.83
*S.* Typhimurium	Control	3.77 ± 0.62 ^a,b^	3.77 ± 0.14 ^c,d^	5.58 ± 0.24 ^a,c^	6.05 ± 0.61 ^b,d^
	Water	3.29 ± 0.42 ^a,b^	3.72 ± 0.45 ^c,d^	4.76 ± 0.44 ^a,c,e^	6.57 ± 0.57 ^b,d,e^
	Liquid plasma	3.47 ± 0.10 ^a,b^	3.24 ± 0.49 ^c,d^	4.40 ± 0.40 ^a,c^	6.10 ± 1.14 ^b,d^
*E. coli*	Control	2.91 ± 0.82	3.53 ± 0.16 ^a^	2.87 ± 0.23 ^a,b^	3.54 ± 0.17 ^b^
	Water	2.89 ± 0.81	3.07 ± 0.33	2.90 ± 0.30	3.36 ± 0.23
	Liquid plasma	2.47 ± 0.98	3.18 ± 0.26	1.96 ± 0.77	2.34 ± 0.59
*C. jejuni*	Control	4.89 ± 0.10 ^a,b,c^	4.62 ± 0.16 ^a,d,e^	6.16 ± 0.37 ^b,d^	7.37 ± 0.02 ^c,e^
	Water	4.44 ± 0.07 ^a^	4.47 ± 0.12 ^b^	5.93 ± 0.22 ^a,b^	7.31 ± 0.14
	Liquid plasma	4.64 ± 0.07 ^a,b^	4.11 ± 0.21 ^c,d^	5.91 ± 0.26 ^a,c^	6.66 ± 1.24 ^b,d^
*S. aureus*	Control	3.60 ± 0.05	3.37 ± 0.36	3.44 ± 0.25	3.38 ± 0.20
	Water	3.21 ± 0.13 ^a^	3.03 ± 0.28	2.92 ± 0.33	2.80 ± 0.28 ^a^
	Liquid plasma	3.17 ± 0.24	3.32 ± 0.32	3.22 ± 0.35	3.05 ± 0.41
*P. aeruginosa*	Control	3.74 ± 0.07 ^a,b^	4.23 ± 0.35 ^c,d^	6.63 ± 0.36 ^a,c^	7.73 ± 0.11 ^b,d^
	Water	3.45 ± 0.35 ^a,b,c^	4.36 ± 0.30 ^a,d,e^	7.5 ± 0.11 ^b,d,f^	7.97 ± 0.02 ^c,e,f^
	Liquid plasma	3.57 ± 0.15 ^a,b^	3.57 ± 0.49 ^c,d^	7.32 ± 0.47 ^a,c,e^	8.01 ± 0.02 ^b,d,e^

^a,b,c,d,e,f^: The same lowercase letters within a row indicate that the values are significantly different at *p* ≤ 0.05.

**Table 5 foods-11-01743-t005:** Averages of the pH value on the surface of the pork samples (*n* = 9).

	Average of pH Value ± SD		
Day	Control	Water	Liquid Plasma
0	5.76 ± 0.09 ^a^	5.66 ± 0.07 ^a^	6.03 ± 0.66 ^a^
3	5.78 ± 0.05 ^a^	5.60 ± 0.02 ^a^	5.70 ± 0.27 ^a^
7	5.66 ± 0.03 ^a^	5.61 ± 0.06 ^b^	5.79 ± 0.39 ^a,b^
10	6.02 ± 0.08 ^a,b^	5.70 ± 0.06 ^a^	5.76 ± 0.38 ^b^

^a,b^: The same lowercase letters within a row indicate that the values are significantly different at *p* ≤ 0.05.

**Table 6 foods-11-01743-t006:** Averages of the pH value of the inside of the pork samples (*n* = 9).

	Average of pH Value ± SD		
Day	Control	Water	Liquid Plasma
0	5.55 ± 0.05 ^a^	5.66 ± 0.06 ^a^	5.98 ± 0.07 ^a^
3	5.68 ± 0.04 ^a^	5.46 ± 0.02 ^a,b^	5.63 ± 0.06 ^b^
7	5.57 ± 0.06 ^a^	5.54 ± 0.04 ^b^	5.68 ± 0.06 ^a,b^
10	5.93 ± 0.07 ^a^	5.54 ± 0.07 ^a^	5.65 ± 0.06 ^a^

^a,b^: The same lowercase letters within a row indicate that the values are significantly different at *p* ≤ 0.05.

**Table 7 foods-11-01743-t007:** Averages of the pH value on the surface of the chicken samples (*n* = 9).

	Average of pH Value ± SD		
Day	Control	Water	Liquid Plasma
0	5.93 ± 0.05 ^a^	6.06 ± 0.09 ^a^	6.22 ± 0.03 ^a^
3	6.00 ± 0.03 ^a^	5.95 ± 0.02 ^a,b^	6.02 ± 0.03 ^b^
7	6.06 ± 0.03 ^a^	5.92 ± 0.07 ^a^	5.99 ± 0.04 ^a^
10	6.13 ± 0.06 ^a^	6.41 ± 0.12 ^a^	6.03 ± 0.03 ^a^

^a,b^: The same lowercase letters within a row indicate that the values are significantly different at *p* ≤ 0.05.

**Table 8 foods-11-01743-t008:** Averages of the pH value of the inside of the chicken samples (*n* = 9).

	Average of pH Value ± SD		
Day	Control	Water	Liquid Plasma
0	5.94 ± 0.04 ^a^	5.99 ± 0.11 ^b^	6.08 ± 0.05 ^a,b^
3	5.95 ± 0.05	5.90 ± 0.03	5.90 ± 0.08
7	5.96 ± 0.05 ^a,b^	5.84 ± 0.06 ^a^	5.86 ± 0.08 ^b^
10	6.05 ± 0.06 ^a^	6.30 ± 0.13 ^a,b^	5.99 ± 0.05 ^b^

^a,b^: The same lowercase letters within a row indicate that the values are significantly different at *p* ≤ 0.05.

**Table 9 foods-11-01743-t009:** Means of color values based on lightness (L*), the red–green index (a*), and the yellow–blue index (b*) on the surface of the pork samples in the control, water immersion, and liquid plasma groups.

Part of Meat		Day 0	Day 3	Day 7	Day 10
	Index		Control Group		
Red meat ^1^	L*	56.26 ± 8.36 ^a^	53.08 ± 12.25	54.99 ± 10.21	43.7 ± 7.52 ^a^
	a*	6.83 ± 2.29	6.29 ± 2.61	6.09 ± 2.39	9.08 ± 1.00
	b*	13.17 ± 3.55	11.69 ± 2.32	10.64 ± 3.26	12.93 ± 3.47
Skin ^1^	L*	58.63 ± 5.79	61.75 ± 4.18	60.05 ± 6.48	60.40 ± 8.29
	a*	7.86 ± 1.45	7.68 ± 1.71	7.58 ± 2.31	5.50 ± 2.25
	b*	17.68 ± 3.40	18.56 ± 2.83	16.50 ± 3.69	15.91 ± 3.90
Side ^2^	L*	57.99 ± 10.71 ^a^	52.74 ± 12.93	37.64 ± 8.69 ^a,b^	51.04 ± 9.17 ^b^
	a*	7.07 ± 1.54	7.87 ± 2.48	8.65 ± 2.30	7.78 ± 1.89
	b*	14.39 ± 1.21 ^a^	14.58 ± 1.51 ^b^	14.00 ± 3.11	16.98 ± 1.61 ^a,b^
	**Water Group**				
Red meat ^1^	L*	45.20 ± 10.10	39.33 ± 4.19 ^a^	43.34 ± 6.90	54.30 ± 6.75 ^a^
	a*	7.52 ± 3.11	8.89 ± 2.46	10.73 ± 2.07 ^a^	6.04 ± 2.80 ^a^
	b*	12.76 ± 3.27 ^a^	9.16 ± 3.66 ^b^	17.17 ± 1.80 ^a,b,c^	11.87 ± 2.76 ^c^
Skin ^1^	L*	65.72 ± 9.25 ^a^	64.71 ± 5.33 ^b^	66.39 ± 2.48 ^c^	76.29 ± 0.79 ^a,b,c^
	a*	3.45 ± 1.18 ^a,b^	6.59 ± 1.49 ^a,c^	7.91 ± 0.84 ^b^	3.26 ± 0.33 ^c^
	b*	16.18 ± 3.25 ^a^	19.18 ± 3.53 ^b^	21.77 ± 1.50 ^a,c^	14.45 ± 0.73 ^b,c^
Side ^2^	L*	58.21 ± 8.89 ^a^	50.94 ± 9.62 ^b^	55.18 ± 9.44 ^c^	68.00 ± 2.77 ^a,b,c^
	a*	5.46 ± 2.67 ^a^	8.19 ± 1.86 ^a,b^	8.16 ± 1.25 ^c^	3.93 ± 0.70 ^b,c^
	b*	14.90 ± 2.41	14.54 ± 1.51 ^a^	17.52 ± 1.91 ^a,b^	13.97 ± 1.58 ^b^
	**Liquid Plasma Group**				
Red meat ^1^	L*	64.10 ± 7.51 ^a^	50.54 ± 7.67 ^a,b^	61.58 ± 6.71	65.91 ± 3.87 ^b^
	a*	5.89 ± 2.12	6.96 ± 1.75	7.88 ± 1.78	6.45 ± 1.66
	b*	13.26 ± 1.42 ^a,b^	12.23 ± 2.42 ^c,d^	17.05 ± 1.97 ^a,c^	16.12 ± 1.42 ^b,d^
Skin ^1^	L*	73.78 ± 1.88 ^a,b,c^	70.19 ± 2.84 ^a,d,e^	78.17 ± 1.38 ^b,d,f^	79.81 ± 0.55 ^c,e,f^
	a*	3.22 ± 0.61 ^a^	5.50 ± 1.70 ^a,b,c^	3.34 ± 0.44 ^b^	3.16 ± 0.30 ^c^
	b*	15.34 ± 0.93 ^a,b^	16.61 ± 2.10 ^c,d^	12.49 ± 1.27 ^a,c,e^	9.27 ± 0.72 ^b,d,e^
Side ^2^	L*	70.26 ± 3.39 ^a,b^	59.07 ± 6.70 ^a,c,d^	71.40 ± 8.30 ^c^	76.62 ± 3.12 ^b,d^
	a*	3.02 ± 0.82 ^a^	6.79 ± 2.19 ^a,b,c^	3.71 ± 0.97 ^b^	3.57 ± 0.49 ^c^
	b*	11.86 ± 0.84 ^a,b^	15.12 ± 2.26 ^a^	13.28 ± 2.28	13.65 ± 0.53 ^b^

^a,b,c,d,e,f^: The same lowercase letters within a row indicate that the values are significantly different at *p* ≤ 0.05; ^1^: n = 9; ^2^: n = 10.

**Table 10 foods-11-01743-t010:** Means of color values based on lightness (L*), the red–green index (a*), and the yellow–blue index (b*) on the surface of chicken meat samples in the control, water immersion, and liquid plasma groups.

Part of Meat		Day 0	Day 3	Day 7	Day 10
	Index		Control Group		
Meat	L*	57.73 ± 6.24 ^a,b^	45.37 ± 5.17 ^a^	48.30 ± 6.81 ^b^	46.96 ± 7.02
	a*	5.11 ± 0.95 ^a^	8.08 ± 1.40 ^a,b^	5.65 ± 1.53 ^b^	5.25 ± 2.55
	b*	16.11 ± 1.42 ^a^	19.63 ± 3.08 ^a^	17.65 ± 3.48	17.17 ± 4.12
Skin	L*	55.40 ± 4.45	59.33 ± 13.36	54.74 ± 10.10	52.07 ± 4.17
	a*	5.66 ± 0.91	5.33 ± 2.32	4.15 ± 1.11	5.99 ± 1.55
	b*	17.15 ± 3.59	16.39 ± 4.67	16.01 ± 3.16	19.02 ± 2.69
	**Water Group**				
Meat	L*	47.06 ± 4.65	49.73 ± 8.49	46.04 ± 10.51	51.47 ± 5.54
	a*	5.49 ± 1.19	7.14 ± 1.88	5.11 ± 1.12	5.38 ± 1.22
	b*	16.74 ± 1.37	18.44 ± 2.15	17.04 ± 2.24	15.98 ± 2.18
Skin	L*	39.15 ± 13.08	50.51 ± 6.46	57.72 ± 9.91	54.73 ± 8.55
	a*	6.52 ± 1.77 ^a^	7.16 ± 1.26 ^b^	3.62 ± 0.89 ^a,b^	4.65 ± 1.50
	b*	16.95 ± 1.82	20.48 ± 2.32 ^a^	16.37 ± 1.99	15.79 ± 2.17 ^a^
	**Liquid Plasma Group**				
Meat	L*	45.89 ± 9.10	48.15 ± 3.12	51.23 ± 5.16	43.18 ± 10.69
	a*	5.08 ± 1.56	6.67 ± 1.24	6.45 ± 2.02	7.10 ± 1.76
	b*	12.16 ± 3.27	19.32 ± 4.71	16.42 ± 3.85	16.40 ± 2.62
Skin	L*	57.89 ± 8.39	61.79 ± 5.05	59.54 ± 7.24	63.14 ± 7.56
	a*	6.06 ± 1.60	5.05 ± 1.70	5.40 ± 0.90	4.56 ± 0.93
	b*	18.74 ± 3.16	17.32 ± 2.75	19.20 ± 3.02	17.24 ± 3.25

^a,b^: The same lowercase letters within a row indicate that the values are significantly different at *p* ≤ 0.05 (n = 9 in each group).

**Table 11 foods-11-01743-t011:** Water activity of pork and chicken meat samples.

Day	Control		Water		Liquid Plasma	
	a_w_	Temperature (°C)	a_w_	Temperature (°C)	a_w_	Temperature (°C)
	Pork Samples					
0	0.990	23.8	0.998	25.0	0.992	25.25
3	0.993	23.7	1.000	25.0	0.984	25.45
7	0.991	24.1	1.002	25.0	0.985	25.45
10	0.991	24.1	1.004	25.0	0.984	25.45
	**Chicken Meat Samples**					
0	0.994	24.8	0.994	24.5	0.994	24.4
3	0.993	24.3	0.994	24.5	0.992	24.4
7	0.994	24.4	0.996	24.3	0.993	24.4
10	0.992	24.1	0.992	24.5	0.994	24.4

## Data Availability

The data presented in this study are available on request from the corresponding author. The data are not publicly available due to institutional privacy policy.

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
