# Peer review of "Decontamination of Pathogenic and Spoilage Bacteria on Pork and Chicken Meat by Liquid Plasma Immersion"

_foods, 2022, doi:10.3390/foods11121743_

Round 1
Reviewer 1 Report
The following paragraph in the Introduction is too repetitive.
The production of high-quality meat and meat products in the food industry faces a major challenge in reducing the risk of pathogens [4]. The global pork output is predicted to increase by 5% from October 2021 to 109.9 million tons in 2022, whereas global poultry meat production is expected to reach 100.8 million tons [5]. Additionally, the availability of pork and poultry meat protein is expected to increase by 13.1% and 17.8%, respectively, by 2030 [2]. Because of increased population growth and the demands of meat consumption, the production of high-quality meat and meat products will also correspondingly increase, which will be a major challenge for the food industry considering the risk of pathogens [4]. Moreover, the production of quality meat and meat products in light of the risk of pathogens will be a great challenge in the food industry [4].
During the slaughtering process, contamination with pathogens including Salmonella spp., Escherichia coli, Campylobacter spp., and spoilage bacteria can also occur [6,7]. This sentence needs to be linked to something else because they say "can also occur" but it is the first time it is mentioned
2.2. Preparation of the Meat. The section needs to specify the muscle (eg. Longissimus dorsi, whole carcass, etc.), also it is important to add more details time between slaughter and purchase, storage temperature, etc. Because in Section 3.3 it is mentioned that is pork belly
2.6.1. Detection and Enumeration of Salmonella (italic)
2.7.1. Meat Color Samples were separately kept in the sterile plastic sealed bags and put in an insulated cooler bag to preserve the temperature at 4–8 °C for the evaluation of meat color, which was performed on days 0, 3, 7, and 10 on the skin, meat and side part with a fat layer (only for the pork sample) using a CIE colorimeter (MiniScan, Hunter Associates Laboratory, Inc., VA, USA) in Figure 1, then the values were recorded as averages [44].
The sentence does not make sense.
Multiple methods of comparison, such as a Bonferroni test with a confidence level of p ≤ 0.05, were used to analyze the significant differences between the mean values of the data from each of the treatment groups. Need to list all of them.
The References need to be checked, they have different format, pay special attention to DOI numbers
Author Response
Response to Reviewer 1 Comments
Point 1: The following paragraph in the Introduction is too repetitive.
The production of high-quality meat and meat products in the food industry faces a major challenge in reducing the risk of pathogens [4]. The global pork output is predicted to increase by 5% from October 2021 to 109.9 million tons in 2022, whereas global poultry meat production is expected to reach 100.8 million tons [5]. Additionally, the availability of pork and poultry meat protein is expected to increase by 13.1% and 17.8%, respectively, by 2030 [2]. Because of increased population growth and the demands of meat consumption, the production of high-quality meat and meat products will also correspondingly increase, which will be a major challenge for the food industry considering the risk of pathogens [4]. Moreover, the production of quality meat and meat products in light of the risk of pathogens will be a great challenge in the food industry [4].
Response 1: Removed “Moreover, the production of quality meat and meat products in light of the risk of pathogens will be a great challenge in the food industry [4].”
Point 2: During the slaughtering process, contamination with pathogens including Salmonella spp., Escherichia coli, Campylobacter spp., and spoilage bacteria can also occur [6,7]. This sentence needs to be linked to something else because they say "can also occur" but it is the first time it is mentioned
Response 2: Revised the sentence to “During the slaughtering process, contamination with pathogens including Salmonella spp., Escherichia coli, Campylobacter spp., and spoilage bacteria can occur [6,7].” in the second paragraph of introduction section.
Point 3: 2.2. Preparation of the Meat. The section needs to specify the muscle (eg. Longissimus dorsi, whole carcass, etc.), also it is important to add more details time between slaughter and purchase, storage temperature, etc. Because in Section 3.3 it is mentioned that is pork belly
Response 3: Meat samples in this study are a pork belly with skin that composed of multiple muscles such cutaneous trunci, latissimus dorsi, pertoralis profundus, rectus abdominis, internal and external abdominal oblique, and fat tissues between these muscles and a chicken breast meat with skin is pectoralis major muscle part.
…The storage temperature of the meat is -18°C.
Point 4: 2.6.1. Detection and Enumeration of Salmonella (italic)
Response 4: (Corrected the word “Salmonella” to italic text.)
Detection and Enumeration of Salmonella
Point 5: 2.7.1. Meat Color Samples were separately kept in the sterile plastic sealed bags and put in an insulated cooler bag to preserve the temperature at 4–8 °C for the evaluation of meat color, which was performed on days 0, 3, 7, and 10 on the skin, meat and side part with a fat layer (only for the pork sample) using a CIE colorimeter (MiniScan, Hunter Associates Laboratory, Inc., VA, USA) in Figure 1, then the values were recorded as averages [48].
The sentence does not make sense.
Response 5: Rearranged sentences as;
2.7.1. Meat Color
The evaluation of meat color, which was performed on days 0, 3, 7, and 10 at the skin site and meat site in both types of meat samples, and side part with a fat layer in the pork sample) using a CIE colorimeter (MiniScan, Hunter Associates Laboratory, Inc., VA, USA) as shown in Figure 1, then the values were recorded the averages.
Point 6: Multiple methods of comparison, such as a Bonferroni test with a confidence level of p ≤ 0.05, were used to analyze the significant differences between the mean values of the data from each of the treatment groups. Need to list all of them.
Response 6: Corrected sentences as;
Multiple comparison as a Bonferroni test with a confidence level of p ≤ 0.05 was used to analyze the significant differences between the mean values of the data from each of the treatment groups.
Point 7: The References need to be checked, they have different format, pay special attention to DOI numbers
Response 7: (Corrected references especially the pattern to show DOI numbers.)
Phan, K.T.K.; Phan, H.T.; Brennan, C.S.; Regenstein, J.M.; Jantanasakulwong, K.; Boonyawan, D.; Phimolsiripol, Y. Gliding arc discharge non-thermal plasma for retardation of mango anthracnose. LWT 2019, 105, 142-148, doi:10.1016/j.lwt.2019.02.012.
Fernández, A.; Thompson, A. The inactivation of Salmonella by cold atmospheric plasma treatment. Food Research International 2012, 45, 678-684, doi:10.1016/j.foodres.2011.04.009.
Ragni, L.; Berardinelli, A.; Vannini, L.; Montanari, C.; Sirri, F.; Guerzoni, M.E.; Guarnieri, A. Non-thermal atmospheric gas plasma device for surface decontamination of shell eggs. Journal of Food Engineering 2010, 100, 125-132, doi:10.1016/j.jfoodeng.2010.03.036.
Kim, H.-J.; Yong, H.I.; Park, S.; Choe, W.; Jo, C. Effects of dielectric barrier discharge plasma on pathogen inactivation and the physicochemical and sensory characteristics of pork loin. Current Applied Physics 2013, 13, 1420-1425, doi:10.1016/j.cap.2013.04.021.
Bolton, D.J. Campylobacter virulence and survival factors. Food Microbiol. 2015, 48, 99-108, doi:10.1016/j.fm.2014.11.017.
Moutiq, R.; Misra, N.N.; Mendonça, A.; Keener, K. In-package decontamination of chicken breast using cold plasma technology: Microbial, quality and storage studies. Meat Science 2020, 159, 107942, doi:10.1016/j.meatsci.2019.107942.

Reviewer 2 Report
Please see attached for extended comments.

Author Response
Response to Reviewer 2 Comments
Point 1: Information on the number of pork/ chicken samples tested or the number replicates to the
experiment was unclear in either the description of the methods or description of statistical
analysis.
Response 1: Added this sentence in 2. Materials and Methods 2.1. Experimental Design:
“Each strain of the bacteria was replicated in seven samples in pork and chicken with skin per group.”
Point 2: Details on when the application of liquid plasma (or water) treatment seemed unclear, were the samples treated with liquid plasma after the 10th day of inoculation, or before storage began?
Were the samples treated four separate times on days 0, 3, 7 and 10? Does the samples with skin
attached performed differently than the ones without?
Response 2: Corrected this sentence in 2.6. Microbiological Analysis:
“Samples were refrigerated in the sterile plastic sealed bags at 4–6 °C to perform the microbial analysis four separate times on days 0, 3, 7, and 10.”
Point 3: Since microbial reduction using liquid plasma was low, considering the concentration of
hydrogen peroxide (H2O2) in liquid plasma used in the study was only 59ppm, Authors should
consider increasing its concentration, perhaps to the legal limit allowed in the slaughtering
process to showcase what the cold plasma is capable of, in terms of reducing microbial loads.
Response 3: Add this point in the fifth paragraph of discussion section:
“Since microbial reduction using liquid plasma was low, considering the higher than 60 ppm hydrogen peroxide (H2O2) in liquid plasma could enhance a decontamination performance. However, a bleaching effect that would be undesirable for the retails and consumers may concern.”
Point 4: From the writings it gave the impression where H2O2 was the active “ingredient” in the liquid plasma, but it was not clear how it differs from hydrogen peroxide, as a form of sanitizer/disinfectant/antiseptic. Would 60ppm of commercially available hydrogen peroxide
yield the same reductions as this series of experiments did?
Response 4: Added the sentence in the first paragraph of the discussion section:
“Gas plasma in free radical forms, such as hydroxyl and hydroperoxyl, is generated under the stimulating energy to excite the gas molecules. These molecules perform charged particles in the form of free radicals. Their microbicidal ability is by interacting with essential cell components and destroying the metabolism of microorganisms [50]. The water droplets in the liquid plasma process can suppress the radical deterioration reaction of hydrogen peroxide. They may enhance plasma efficiency [51]. The argon plasma is one of the most promising choices for producing a high concentration of H2O2 since the hydroxyl dissolved in the argon plasma and hydroxyl radicals are the principal source of H2O2 formation that will be transfer to treated liquid or PAW and contribute to the inactivation of bacteria [52].”
Point 5: For the storage study, including aerobic plate count throughout sampling period would provide a better description on microbial loads on the meat samples. A negative control, where the meat samples weren’t inoculated with any strains seemed to be missing.
Response 5: We performed mesophilic aerobic plate count but were not complete for all data. This study aims to focus on each bacteria strain that was counted on the selective media. The uninoculated chicken meat were not found salmonella and campylobacter in 25 gram (n=5).
Point 6: Statistical analysis between bacterial strains or between meat products seems to be excluded,
authors should determine if deducing a statistical conclusion of the treatment on different microbial strains or meat products is important enough for the manuscript.
Response 6: At first time before the authors stated to design the experiment, the statistical analysis between meat products was one of objectives that we would like to analyse. However, as the microbial loads results at each types of bacteria were relatively by each day of experiment and also one of factors that relatively concerned and deminished bacterial inactivation is the surface convolution of the chicken with skin sample which is naturally rough more than the surface of the pork with skin sample resulted in focusing on statistical conclusion of the treatements in each types of meat.
Revised to
The reduction of C. jejuni was reduced significantly when compared with the reduction of S. Typhimurium within the liquid plasma group (p ≤ 0.05) as shown in Table 1. However, other bacterial reduction results in water and plasma treatments were not significantly different in pork and chicken samples (Table 1 and Table 3).
Point 7: A large portion of the discussion was attributed to the color measurements, but there seem to be missing justifications on the usage of color space to determine the quality of the meat samples
during storage, or as an indicator on the microbial load, or whatever the authors intended.
Response 7: Added this reference in the seventh paragrph of discussion section:
“Color and overall appearance of meat have a greater effect on customer acceptance before consumption than flavor and texture [68].”
Point 8: “Global meat protein consumption is expected to rise by 14% by 2030, compared with the period average during 2018-2020”
Missing base number for comparison.
Response 8: Corrected this sentence in the first parapraph of introduction section to:
“Global meat protein consumption is expected to rise by 14% at an estimated 138 million tons by 2030, compared with the period average of 121 million tons during 2018–2020 [2].”
Point 9: “Additionally, the health and safety of meat handlers should be considered, including the risk of resistant acid bacteria,”
Resistant acid bacteria? What does that mean?
Response 9: Resistant acid bacteria in the references means E. coli O157:H7. Thus the authors corrected the sentence in the third paragraph of introduction section to “Additionally, the health and safety of meat handlers should be considered, including the risk of resistant acid bacteria such as E. coli O157:H7 [25],”
Point 10: “carcinogenic compounds from working with chlorine”
What are some of the carcinogenic chlorine by-products?
Response 10: Added the carcinogenic chlorine by-products mentioned in the reference which is trihalomethanes (THMs) in the third paragraph of introduction section:
“the risk of carcinogenic compounds as trihalomethanes (THMs) from working with chlorine [18,28]”
Point 11: “a novel decontamination technology that has the potential to aid in the elimination of pathogens in food or meat products and has been used in the meat business to overcome the limits of existing decontamination methods.”
What are the existing decontamination methods? And what are their limits?
Response 11: The limits of existing decontamination methods from the context that the authors intended to write is the limitations of conventional decontamination methods that mentioned before in the manuscript.
Therefore, the authors corrected the sentence to “the limitations of conventional decontamination methods as previously mentioned.” in the third paragraph of introduction section.
Point 12: “plasma can emit reactive species such as gas and oxygen compounds,”
What does “gas” here mean?
Response 12: The word “gas” in the sentence the authors means reactive oxygen and nitrogen species. Therefore, the authors corrected the sentence to “plasma can emit reactive oxygen and nitrogen species” in the third paragraph of introduction section.
Point 13: “Reducing the amount of bacterial contamination during the slaughtering process is a crucial step for the control point.”
Is this the only CCP available during the slaughtering process? Where else can bacterial contamination be controlled?
Response 13: At first time, the authors intended to write that Reducing the amount of bacterial contamination during the slaughtering process is one of important steps for the control point.
Thus, we corrected this sentence in the fourth paragraph of introduction section to show the control point of each types of meat which relate to our study:
“Reducing the amount of bacterial contamination during the slaughtering process such as chilling in poultry or final washing in pig are categorized as crucial steps for the control point.”
Point 14: “In the liquid plasma group, meat samples were immersed in 500 mL of 60 ppm of H2O2 for 15 min at 25.5 °C”
How does liquid plasma H2O2 differ from normal H2O2 found in grocery stores?
Response 14: -same in point 4-
Point 15: “A total of 104 CFU/mL of the bacterial suspension was inoculated”
Inconsistent notation, use log CFU/mL as shown in section 2.2, and 2.8.
How was the consistent 4 log CFU/mL inoculation on all meat samples ensured?
Response 15: Corrected to “4 log CFU/mL”
Bacteria concentration was measureed at 0.5 MacFarland by densitometer. It contains approximately 108 or 8.0 Log CFU per mL. Then, ten-fold dilution was done to the final concentration at 104 or 4.0 Log CFU per mL. The bacterial number of final inoculation was checked by the drop plating method on selective media for all batces of the bacterial suspension.
Point 16: “The cutting size of the meat was approximately 15 × 15 × 4 cm.”
What cut of meat from chicken and pork is used in the study? Pork belly? Chicken breast?
Response 16: Added this sentence in 2.2. Preparation of the Meat:
“Meat samples in this study are a pork belly with skin that composed of multiple muscles such cutaneous trunci, latissimus dorsi, pertoralis profundus, rectus abdominis, internal and external abdominal oblique, and fat tissues between these muscles [33] and a chicken breast meat with skin is pectoralis major muscle part.”
Point 17: “A 25 g meat sample was mixed with 225 mL Bolton broth”
“The sample with skin was cut to 25 g and placed in a sterile stomacher bag,”
Why are some samples with skin on and some samples with skin off? Are the skin-on
samples exclusive to chicken?
Response 17: All the words “meat sample” or “sample” appeared in this manuscript mean meat sample with skin. Therefore, the authors corrected all the word “sample with skin” to “meat sample” or “sample” and specified the section of pork and chicken samples in 2.2. Preparation of the Meat.
Point 18: “15 min with the artificial contamination of S. Enteritidis, S. Typhimurium, E. coli, and C.
jejuni”
Remind readers the initial inoculation load.
Suggest changing “artificial contamination” to “inoculation”
Response 18: Corrected all the word “artificial contamination” in the manuscript to “inoculation”
Point 19: “Reduction is the difference of the mean (log CFU/g) of the bacteria between the initial load and each treatment.”
Hard to follow what the initial inoculation load without it stated in the respective section.
Response 19: The authors added initial loads (Control) in Table 1 and Table 3.
Point 20: “In the decontamination experiment, the application of liquid plasma with 60 ppm of H2O2 for 15 min significantly reduced the microbial loads of S. Enteritidis, S. Typhimurium, and E. coli of
the chicken meat's surface”
Are there supporting stats? Are there missing differential indicators in Tables 1 & 3?
Response 20: At first time, the authors wrote this sentence by refering from the significance at each time point that was illustrated in Table 4 which is relatively not clearly spefied to the overall significance of each microbial load results for reader as the reviewer’s comment, thus the authors corrected this sentecne to “In the decontamination experiment, the application of liquid plasma with 60 ppm of H2O2 for 15 min could reduce the microbial loads of S. Enteritidis, S. Typhimurium, and E. coli of the chicken meat's surface” in the manuscript. And the authors recently added differential indicators which shown in Table1 (Table 3 was no significant diffences).
Point 21: “In our trials, the artificial contamination had a bacterial concentration of 4 log CFU/mL, which was a relatively high level of initial load. As a result, this has the potential to compromise the
decontamination efficiency of liquid plasma.”
Missing discussion on why water treatment performed similarly to the liquid plasma treatment.
Response 21: Corrected the sentence in fifth paragraph in discussion section:
“In our trials, the artificial inoculation had a bacterial concentration of 4 log CFU/mL, which was a relatively high level of initial load. Furthermore, maintaining stability of reactive oxygen species such as hydrogen peroxide while storage is essential to antimicrobial properties, the decrease of hydrogen peroxide over storage time at 25 °C, 4 °C, and -20 °C has been suggested to affect microbial inactivation by PAW compared to PAW stored at -80 °C [60]. In addition, our experiment was performed at the room temperature (25–30 °C) affecting the antimicrobial activity of liquid plasma as well as the H2O2 concentration could be decreased over period of time. Since microbial reduction using liquid plasma was low, considering the higher than 60 ppm hydrogen peroxide (H2O2) in liquid plasma could enhance a decontamination performance. However, a bleaching effect that would be undesirable for the retails and consumers may concern [65]. As the results, the reducing antimicrobial activity that might lead to the liquid plasma treatment performed similarly to the water treatment. Therefore, that have the potential to compromise the decontamination efficiency of liquid plasma. A combination of liquid plasma treatment and good hygienic practices in the slaughtering process could be more efficient in lowering the bacterial load in meat.”
